# Comparison of Bioluminescent Substrates in Natural Infection Models of Neglected Parasitic Diseases

**DOI:** 10.3390/ijms232416074

**Published:** 2022-12-16

**Authors:** Sarah Hendrickx, Dimitri Bulté, Dorien Mabille, Roxanne Mols, Mathieu Claes, Kayhan Ilbeigi, Rokaya Ahmad, Laura Dirkx, Sara I. Van Acker, Guy Caljon

**Affiliations:** Laboratory of Microbiology, Parasitology and Hygiene (LMPH), University of Antwerp, 2610 Wilrijk, Belgium

**Keywords:** reporter gene technology, bioluminescence, D-luciferin, CycLuc1, AkaLumine-HCl, *Trypanosoma*, *Leishmania*

## Abstract

The application of in vivo bioluminescent imaging in infectious disease research has significantly increased over the past years. The detection of transgenic parasites expressing wildtype firefly luciferase is however hampered by a relatively low and heterogeneous tissue penetrating capacity of emitted light. Solutions are sought by using codon-optimized red-shifted luciferases that yield higher expression levels and produce relatively more red or near-infrared light, or by using modified bioluminescent substrates with enhanced cell permeability and improved luminogenic or pharmacokinetic properties. In this study, the in vitro and in vivo efficacy of two modified bioluminescent substrates, CycLuc1 and AkaLumine-HCl, were compared with that of D-luciferin as a gold standard. Comparisons were made in experimental and insect-transmitted animal models of leishmaniasis (caused by intracellular *Leishmania* species) and African trypanosomiasis (caused by extracellular *Trypanosoma* species), using parasite strains expressing the red-shifted firefly luciferase PpyRE9. Although the luminogenic properties of AkaLumine-HCl and D-luciferin for in vitro parasite detection were comparable at equal substrate concentrations, AkaLumine-HCl proved to be unsuitable for in vivo infection follow-up due to high background signals in the liver. CycLuc1 presented a higher in vitro luminescence compared to the other substrates and proved to be highly efficacious in vivo, even at a 20-fold lower dose than D-luciferin. This efficacy was consistent across infections with the herein included intracellular and extracellular parasitic organisms. It can be concluded that CycLuc1 is an excellent and broadly applicable alternative for D-luciferin, requiring significantly lower doses for in vivo bioluminescent imaging in rodent models of leishmaniasis and African trypanosomiasis.

## 1. Introduction

Studying the kinetics of protozoan infections has been greatly enhanced by advances in imaging techniques and the introduction of reporter gene technology [1,2]. Reporter genes encode proteins with readily measurable phenotypes that are easily distinguished over the endogenous protein background. Bioluminescent imaging (BLI) using firefly luciferase is a powerful and non-invasive tool to study infectivity as well as molecular and cellular features in live animals. Firefly luciferase produces light through the enzymatic oxidation of D-luciferin into oxyluciferin in the presence of ATP, oxygen and Mg^2+^ [3,4,5]. The emitted photons can subsequently be detected by various in vitro and in vivo bioluminescence systems. One of the major hurdles for in vivo BLI infection models is that microbial pathogens expressing the firefly luciferase enzyme do not endogenously produce D-luciferin, requiring administration of this substrate prior to imaging. Although D-luciferin has been widely used for BLI, its relatively low wavelength emission spectrum (λ_max_ = 562 nm) in combination with the firefly luciferase Luc2 is believed to make this enzyme–substrate pair less suitable for deep tissue imaging due to absorption by hemoglobin (λ = 415 − 577 nm) and melanin (λ < 600 nm) [6,7]. Additionally, D-luciferin has a low permeability for certain tissues, such as the brain [8], resulting in a heterogeneous biodistribution [9,10]. These limitations can be partially overcome by using codon-optimized red-shifted luciferases that are well expressed and emit red or near-infrared light, which is less absorbed by the host tissue, resulting in an overall better tissue penetration [6,11,12]. PpyRE9 is an example of such a red-shifted luciferase that has been developed by introducing specific amino acid changes in the *Photinus pyralis* luciferase, resulting in a relative shift of light emission to a >600 nm wavelength [13]. Another way to overcome the poor tissue penetration of emitted light and the heterogeneous biodistribution of D-luciferin is to use modified bioluminescent substrates with enhanced cell permeability, improved luminogenic and pharmacokinetic properties [14].

CycLuc1 (λ_max_ = 604 nm) and AkaLumine-HCl (λ_max_ = 677 nm) are two synthetic D-luciferin analogues with a near-infrared emission spectrum, hereby improving in vivo characteristics such as tissue penetration and distribution [15]. These enhanced in vivo characteristics result from the addition of a cyclic alkyl amino group, in case of CycLuc1, and the substitution of the benzothiazole moiety by an aromatic group, in case of AkaLumine-HCl [15,16,17]. Bioluminescent imaging of 4T1-luc2 tumor xenografts with CycLuc1 yielded a more than 10-fold higher bioluminescent signal compared to D-luciferin at equivalent doses [15]. Furthermore, CycLuc1 was shown to cross the blood–brain barrier more effectively, resulting in improved bioluminescent signals from the brain [15]. AkaLumine-HCl, on the other hand, appeared to have a higher in vitro bioluminescent output compared to CycLuc1 and D-luciferin (>6.7 fold higher), with the maximal signals recorded at lower concentrations [17]. In vivo, AkaLumine-HCl showed a similar bioluminescent output as CycLuc1 for the imaging of subcutaneous tumors [17]. Moreover, bioluminescent signals arising from lung metastasis using AkaLumine were up to 8-fold higher compared to D-luciferin and 3-fold higher compared to CycLuc1 [17].

Next to extensive applications in cancer research, BLI has proven to be a valuable research tool in the field of infectious parasitic diseases [2]. The generation of bioluminescent parasites not only enables the evaluation of in vivo infection kinetics [18,19,20,21,22,23], treatment dynamics [22,24,25,26,27,28,29,30,31], and parasite dissemination upon natural transmission, but can also support the search for hidden parasite niches [32,33,34]. For leishmaniasis and trypanosomiasis, two closely-related neglected parasitic diseases, several research groups, including ours, have already successfully used BLI for the abovementioned applications. Although multiple luciferase–substrate systems have been explored for trypanosomiasis and leishmaniasis, combination of a red-shifted luciferase and D-luciferin was proven to be suitable for in vivo application [35,36]. In this study, two recently modified bioluminescent substrates, CycLuc1 and AkaLumine-HCl, are compared to D-luciferin for in vitro and in vivo application in models of cutaneous and visceral leishmaniasis (CL and VL) and African trypanosomiasis (AT).

## 2. Results

### 2.1. CycLuc1 Has a Moderately Higher Potency to Detect Parasites In Vitro

Comparison of different substrates for the detection of PpyRE9-expressing parasites revealed that they are all efficacious in vitro (Figure 1). CycLuc1 demonstrated a higher luminescent signal than D-luciferin and AkaLumine-HCl, both for extracellular (*L. infantum* and *L. major* promastigotes and *T. brucei*) and intracellular parasites (intramacrophage *L. infantum* and *L. major*). This effect was apparent for *L. infantum* promastigotes for substrate concentrations ranging from 250 µM to 2.5 µM (F_(6, 24)_ = 945.4, *p* < 0.0001; Figure 1A), but only for the highest concentration tested against *L. major* promastigotes (F_(6, 60)_ = 74.1, *p* < 0.0001; Figure 1B). Moreover, although the same effect could be observed for the highest concentration tested against intracellular *L. infantum* amastigotes (Figure 1C) (F_(6, 60)_ = 5.018, *p* = 0.0003), no statistical differences could be demonstrated for *L. major* amastigotes (Figure 1D).

When comparing the efficacy of the substrates against *T. brucei*, the addition of CycLuc1 also resulted in higher luminescence (Figure 1E), albeit only statistically significant at the two highest concentrations tested (F_(6, 60)_ = 46.04, *p* < 0.0001). Given the comparable in vitro efficacy, all substrates were also tested in subsequent in vivo experiments.

### 2.2. CycLuc1 Requires Lower Doses for Sensitive Leishmania Infantum Detection in Dermis and Visceral Organs

In a next stage, the different substrates were evaluated in in vivo models of dermal and visceral *Leishmania* infection. As such, the performance of the substrates can be compared in the superficial skin and in the main visceral organs. As naïve, non-infected mice showed extremely high background signals in the liver upon injection of AkaLumine-HCl (Appendix A), this substrate revealed to be unsuitable for straightforward application in infectious disease research and was excluded from further in vivo comparisons. When the LoD of hepatic *Leishmania* infection was compared between D-luciferin and CycLuc1, the latter showed a comparable efficacy at a 20-fold lower dose (150 mg/kg versus 7.5 mg/kg) (Figure 2A,B) (U = 0, *p* = 0.0238). Both substrates can be used to detect parasites levels above 10^6^ parasites. When comparing the LoD between D-luciferin at 150 mg/kg and CycLuc1 at 7.5 mg/kg upon i.v. and i.d. infection, no significant differences could be observed (Figure 2C,D).

### 2.3. CycLuc1 Requires Lower Doses for Detection of Naturally Transmitted Leishmania and Trypanosoma Infections

Considering sand fly transmitted *L. major* infections, strong BLI signals could be detected at the infection site (ear) following administration of 150 mg/kg D-luciferin or 7.5 mg/kg CycLuc1. Although significantly higher levels of bioluminescence were recorded upon use of D-luciferin (Figure 3A,B) (F_(5, 12)_ = 5.109, *p* = 0.0097), both substrates were highly effective. Comparable trends were obtained when transmission occurred either on the ear or on the rump (Appendix A). The observed increase in bioluminescent signal over the course of infection seemed to correspond to the visual aggravation of dermal pathology as well (Figure 3C). For sand fly transmitted *L. infantum* LLM-2346*^PpyRE9^* parasites, no signal could be detected in either the ear or visceral organs with both substrates (Appendix A), indicating that the transmitted parasite inoculum was probably below the LoD of the in vivo imaging system.

When comparing D-luciferin and CycLuc1 in tsetse transmitted *T. brucei* AnTAR1*^PpyRE9^* infections in mice, both substrates were effective for longitudinal assessment of infection. No significant differences in bioluminescence were recorded from ventral and dorsal sides over the entire course of infection (Figure 3D–F).

## 3. Discussion

During the last decade, BLI techniques in parasitology have been gaining momentum as they represent a fast, reliable and more ethical approach for the longitudinal and non-invasive follow-up of in vivo infections. This approach facilitates the evaluation of infection dynamics and treatment efficacy without the need to euthanize groups of mice at specific time points. Various enzymes, derived from marine organisms (*Gaussia*, *Oplophorus* and *Renilla*) and insects (click beetle, firefly), have found ubiquitous use as bioluminescent reporters. In the field of parasitology, firefly luciferase transfected parasites are generally combined with D-luciferin administration. However, the relatively low wavelength emission spectrum associated with absorption by hemoglobin and melanin and heterogeneous biodistribution due to a low permeability of D-luciferin for certain tissues should be considered [6,7]. That is why, in the last years, modified red-shifted luciferases and modified bioluminescent substrates with enhanced tissue penetration characteristics and improved luminogenic properties have gained interest [6,11,12]. For in vivo *Leishmania* and *Trypanosoma* imaging, combination of a red-shifted luciferase transgene with D-luciferin substrate has proven to be effective [35,36]. In this study, two modified bioluminescent luciferase substrates, CycLuc1 and AkaLumine-HCl, were compared with D-luciferin for in vivo detection of PpyRE9-expressing parasites in animal models of VL, CL, and AT.

In a recent in vitro comparison between different near-infrared substrates in human HEK93T cells, AkaLumine-HCl was elected as one of the best substrates [37]. At very low concentrations, AkaLumine-HCl was also reported to have a considerably enhanced sensitivity for in vivo imaging of deep-tissue targets compared to CycLuc1 and D-luciferin [17]. Moreover, its long half-life in serum (~40 min), causing extended BLI signals, endorsed in vivo application [17]. AkaLumine-HCl could successfully be used to detect subcutaneous tumors and lung metastases with an over 40-fold higher signal to that obtained by D-luciferin at equal intraperitoneal doses [17].

Previous research on cancer cells showed that D-luciferin and CycLuc1 caused a dose-dependent increase in signal and were weak at low concentrations, whereas the efficacy of AkaLumine-HCl was less concentration-dependent, generating a strong signal even at low concentrations. This effect has been attributed to an increased cell-membrane permeability [17]. However, our in vitro results clearly demonstrate a dose-dependency of AkaLumine-HCl for *Leishmania* and *Trypanosoma* spp. In addition, the in vitro signal generated by AkaLumine-HCl was not higher than those of D-luciferin and CycLuc1, suggesting that permeability of the substrates for parasites and cancer cells are different. Despite the promising in vivo results obtained by other research groups, the in vivo application of AkaLumine-HCl in our study to monitor VL, CL and AT was hampered by the high hepatic bioluminescent background signal even at low doses (50 mg/kg ~3 µM). High background signals have recently also been reported by another group [38] and triggered the exploration of other substrate derivatives [39]. The hepatic background signal clearly compromised the use of Akalumine-HCl. The liver is a major target organ for *Leishmania*, and the amount of parasites needed to generate a detectable BLI signal was rather high (LoD of 10^5^–10^7^) [31]. Moreover, it has been shown previously that the intracellular amastigote form of *Leishmania* is transcriptionally and metabolically less active, which would result in a lowered production of light [31]. CycLuc1 showed an enhanced efficacy over D-luciferin at doses that were 20-fold lower. Upon comparison of low doses of CycLuc1 (7.5 mg/kg) with the standard dose of D-luciferin (150 mg/kg), similar bioluminescent signals and LoD levels were detected in all tested infection models, proving that CycLuc1 can indeed be administered at lower doses in all the evaluated models.

## 4. Materials and Methods

### 4.1. Animals

Six to eight week-old, female BALB/c and Swiss mice were purchased from Janvier Laboratories (Le Genest-Saint-Isle, France) for the in vivo infection studies and the collection of peritoneal macrophages, respectively. Mice were randomly allocated to experimental groups and housed in individually ventilated cages with access to laboratory rodent food (Carfil, Oud-Turnhout, Belgium) and drinking water ad libitum.

### 4.2. Parasites and Transfections

For the experiments with *Leishmania* parasites, *L. infantum* LLM-2346 (MHOM/ES/2016/LLM2346; a kind gift of Dr. Javier Moreno of the Instituto de Salud Carlos III in Madrid) and *L. major* JISH118 (MHOM/SA/85/JISH118; a kind gift of Dr. Simon Croft of the London School of Hygiene and Tropical Medicine in London) promastigotes were transfected with the red-shifted firefly luciferase variant *PpyRE9* that was codon optimized for expression in *Leishmania* (Genscript biotech, Rijswijk, The Netherlands). The *PpyRE9* gene was integrated into the pLEXSY-hyg2.0 expression vector (Jena Bioscience, Jena, Germany) using the NcoI and NotI restriction sites. The pLEXSY-hyg2.0-PpyRE9 vector was subsequently digested using SwaI and the 6.8 kb fragment was gel-purified using the QIAquick Gel Extraction Kit (Qiagen Benelux b.v., Venlo, The Netherlands), followed by ethanol precipitation and resuspension in cytomix transfection buffer (120 mM KCl; 0.15 mM CaCl_2_; 10 mM KH_2_PO_4_; 25 mM HEPES; 2 mM EDTA; 5 mM MgCl_2_; pH 7.6) at 1 µg/µL. Ten µg of the linearized construct was used for the electroporation of 1 × 10^8^ procyclic promastigotes (twice at 25 µF and 1500 V with 10 s interval) using the Bio-Rad GenePulse Xcell electroporation unit (Bio-rad Laboratories, Hercules, CA, USA). Transfectants were selected under hygromycin (Jena Bioscience, Jena, Germany) pressure (100 µg/mL) and subcultured twice a week in HOMEM promastigote medium (Invitrogen, Waltham, MA, USA) supplemented with 10% heat inactivated fetal bovine serum (iFBS). To maintain their virulence, parasites were maintained in vitro for a maximum of 10 passages and were repeatedly passaged in vivo. For the African trypanosome infections, pleomorphic post-fly *Trypanosoma brucei brucei* AnTAR1*^PpyRE9^* parasites were used [36,40], which were collected from the tail vein of infected donor mice.

### 4.3. Infection of Sand Flies and Tsetse Flies for Transmission Studies

*Lutzomyia longipalpis* sand flies were maintained under standard conditions [41]. Logarithmic phase *Leishmania* LLM-2346*^PpyRE9^* and *L. major* JISH118*^PpyRE9^* promastigotes were used to infect sand flies by feeding two hundred 5–7 day-old female sand flies an infected blood meal containing 5 × 10^6^ promastigotes/mL as described earlier [42,43,44]. Twenty-four hours post infection, blood meal engorged females were separated and maintained until days 9 (*L. infantum*) and 7 post infection (*L. major*), respectively. At that moment, a minimum of 10 sand flies were dissected and their midgut was evaluated microscopically for the presence of metacyclic parasites colonizing the fly’s stomodeal valve, being an indicator for successful transmission.

Tsetse fly pupae were imported from Bratislava via a collaboration with Dr. Peter Takáč (Slovak Academy of Sciences, Slovakia). *T. b. brucei* AnTAR1*^PpyRE9^* parasites were used to infect tsetse flies (*Glossina morsitans morsitans*) by feeding newly emerged flies with an infected blood meal supplemented with 10 mM reduced L-glutathione. The infected blood meal consisted of a mixture of parasitized blood from infected mice at day 5–7 post infection and defibrinated horse blood to obtain a concentration of >10^6^ blood stream form parasites/mL. After this infected blood meal, tsetse flies were fed every 2–3 days with defibrinated horse blood. Four weeks post infection, tsetse flies were allowed to probe on pre-heated glass slides, which were then microscopically examined for the presence of metacyclic trypanosomes. Flies demonstrating the presence of metacyclic parasites were used for onward transmission studies.

### 4.4. Comparison of Different Substrates for Parasite Detection In Vitro

The light-producing capacity of the transfected *L. infantum* LLM-2346*^PpyRE9^* and *L. major* JISH118 *^PpyRE9^* promastigotes and *T. brucei brucei* AnTAR1*^PpyRE9^* parasites was evaluated first in vitro. Log phase *L. infantum* LLM-2346*^PpyRE9^* or *L. major* JISH118 *^PpyRE9^* promastigotes were seeded in black 96-well plates with a clear bottom at a final concentration of 500,000 parasites/75 µL well. *T. brucei* parasites, collected from the blood of heavily infected donor mice, were plated at a final concentration of 100,000 parasites/75 µL well. Immediately after seeding the parasites, 25 µL of D-luciferin (beetle luciferin potassium salt, Promega, Madison, WI, USA), CycLuc1 (Merck Life Science B.V., Amsterdam, The Netherlands) and AkaLumine-HCl (TokeOni; Merck Life Science B.V., Amsterdam, The Netherlands) (dissolved in PBS at 30 mg/mL, 1.5 mg/mL and 10 mg/mL respectively) were added to the plate at concentrations ranging from 250 µM to 0.25 µM. The luminescence [in relative luminescence units (RLU) p/s/cm^2^/Sr] was measured using the GloMax Explorer (Promega Benelux, Leiden, The Netherlands).

As *Leishmania* is an intracellular parasite, in vitro intracellular bioluminescence was determined in parallel. For this purpose, peritoneal macrophages were collected from starch-stimulated Swiss mice as described earlier [45]. Macrophages were plated in black 96-well plates with a clear bottom at a final concentration of 30,000 cells/100 µL. Twenty-four hours later, macrophages were infected with late stationary phase *L. infantum* LLM-2346*^PpyRE9^* or *L. major* JISH118^PPyRE9^ promastigotes at a ratio of 15:1. Ninety-six hours post infection, the medium was discarded and cells were washed twice with heat-inactivated horse serum (Invitrogen, Waltham, MA, USA) to remove extracellular promastigotes [46]. Then, 75 µL of fresh culture medium was added together with 25 µL of the different substrates at similar concentrations as described above. Intracellular bioluminescence was measured using the GloMax Explorer (Promega Benelux, Leiden, The Netherlands). An infected control plate was fixed with methanol and stained with Giemsa in parallel to confirm intracellular infection.

### 4.5. Determination of the Detection Limit of the Different Substrates in the In Vivo L. Infantum Model

BALB/c mice were infected either intravenously (i.v.) or intradermally (i.d.) with a 1:10 dilution series of *L. infantum* LLM-2346*^PpyRE9^* promastigotes. Based on our previous findings regarding the limit of detection (LoD) in the BLI model of VL [31], a range from 10^5^ to 10^8^ or from 10^3^ to 10^5^ parasites were used for the i.v. and i.d. infections, respectively. Parasite burdens in inoculation sites and the main target organs were determined with the IVIS Spectrum (PerkinElmer, Brussels, Belgium). One hour post infection (hpi) and 1 day post infection (dpi) mice were anesthetised via 2% isoflurane inhalation and imaged at for 15 min starting 3 min after i.p. injection of 150 mg/kg D-luciferin (Beetle Luciferin Potassium Salt, Promega Benelux, Leiden, The Netherlands) or 7.5 mg/kg CycLuc1 (Merck Life Science B.V., Amsterdam, The Nederlands) or 50 mg/kg AkaLumine-HCl (Merck Life Science B.V., Amsterdam, The Nederlands). Substrate concentrations were selected based on earlier in vivo studies in mice [15,47,48]. Images were analyzed using LivingImage v4.3.1 software (PerkinElmer, Brussels, Belgium) by quantifying the signal as relative luminescence units (RLU) within regions of interest corresponding to the ear, liver, spleen and bone marrow. To compare potential background signals, naïve, non-infected mice were imaged in parallel.

### 4.6. Comparison of Different Substrates for Infection Follow-Up in Mice

To evaluate the in vivo application of the different substrates, mice were either infected with *L. infantum* LLM-2346*^PpyRE9^*, *L. major* JISH118*^PPyRE9^* or *T. brucei* AnTAR1*^PpyRE9^* via natural insect-mediated transmission or with *L. infantum* LLM-2346*^PpyRE9^* by i.v. injection of 1 × 10^8^ late stationary phase promastigotes. To initiate natural transmissions, Swiss (*Trypanosoma*) or BALB/c mice (*Leishmania)* were anaesthetized with a 87.5 mg/kg xylazine/12.5 mg/kg ketamine mixture, after which one *T. brucei* AnTAR1*^PpyRE9^* infected tsetse fly (*Glossina morsitans morsitans*) or twenty *Leishmania* infected *Lu. longipalpis* sand flies were allowed to feed on the right ear. In vivo parasite burdens were monitored over time throughout early (1 to 3 days post infection for *Leishmania* and *Trypanosoma*, respectively) or late infection (up to 12 days for *Trypanosoma* and up to 5 weeks for *Leishmania*) using the IVIS Spectrum (PerkinElmer, Waltham, MA, USA). Mice were imaged for 10 min (*Leishmania*) or 1 s (*Trypanosoma*) starting 3 min after i.p. injection of 150 mg/kg D-luciferin or 7.5 mg/kg CycLuc1 and were kept under 2–2.5% isoflurane anesthesia during the measurements. Mice were imaged from their ventral or dorsal sides, depending on the location of the regions of interest (ROI). Images were analyzed using LivingImage v4.3.1 software (PerkinElmer, Waltham, MA, USA) by quantifying the signal as relative luminescence units (RLU) within ROIs corresponding to the ear (*L. infantum, L. major* and *T. brucei* after transmission) or liver, spleen, and bone marrow (*L. infantum* after i.v. infection). Specificity of the signal at the ROIs was checked by comparing images to those of non-infected control animals.

### 4.7. Statistics

Statistical differences between groups were evaluated using Graphpad Prism 9 software. Statistical differences in in vitro BLI signal were evaluated using a 2-way ANOVA with Tukey’s multiple comparisons test. Comparison of the LoD of the different substrate concentrations was performed using a Mann–Whitney U test. To evaluate potential differences between the BLI signal emitted by infected animals, a 2-way ANOVA was used with a Šídák’s multiple comparisons test.

## 5. Conclusions

Our data collectively demonstrated the broad applicability of CycLuc1 in natural infection models of the major neglected tropical diseases leishmaniasis and African typanosomiasis. While AkaLumine-HCl cannot be used in these models, CycLuc1 is equally effective as D-luciferin at 20-fold lower doses, and therefore represents an excellent alternative for longitudinal follow-up of these parasitic infections in mice. Given the commercial availability of both CycLuc1 and D-luciferin, differences in the dosing regimen can be included in cost-benefit analyses.

## Figures and Tables

**Figure 1 ijms-23-16074-f001:**
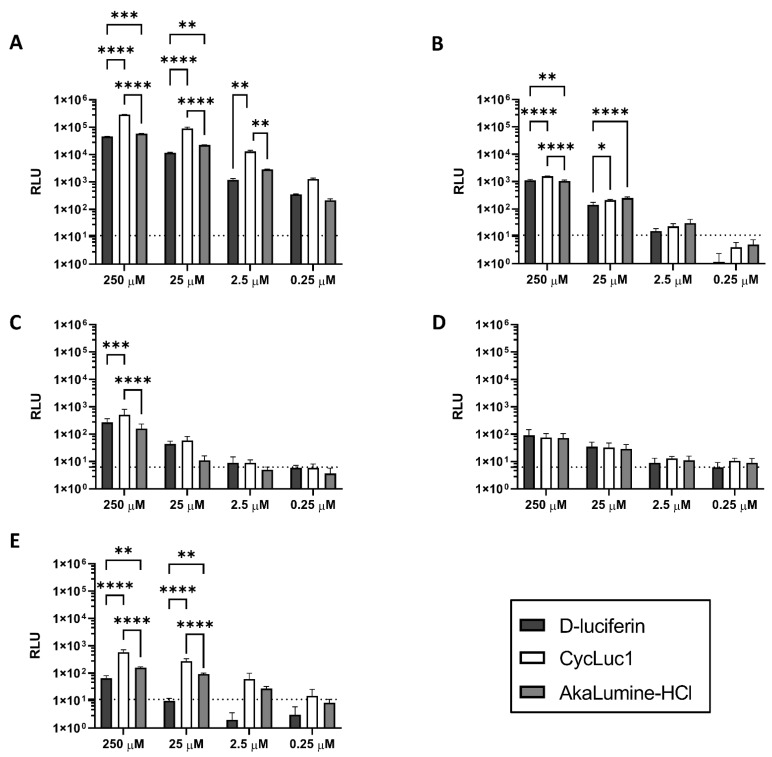
In vitro bioluminescence of different concentrations of D-luciferin, CycLuc1 and AkaLumine-HCl. (**A**) In vitro bioluminescent signal of *L. infantum* LLM-2346*^PpyRE9^* promastigotes and (**B**) *L. major* JISH118*^PpyRE9^* promastigotes; (**C**) In vitro bioluminescent signal of *L. infantum* LLM-2346*^PpyRE9^* amastigote-infected mouse peritoneal macrophages and (**D**) *L. major* JISH118*^PpyRE9^* amastigote-infected mouse peritoneal macrophages; (**E**) In vitro bioluminescent signal of *T. brucei brucei* AnTAR1.1*^PpyRE9^* parasites. The representative figures show the average luminescence (expressed in relative luminescence units (RLU) [p/s/cm^2^/Sr]) produced in at least two independent repeat experiments, each conducted in triplicate. Dotted lines represent the background signal. Results are expressed as mean luminescent signal ± SD (* *p* ≤ 0.05, ** *p* ≤ 0.01, *** *p* ≤ 0.001 and **** *p* ≤ 0.0001).

**Figure 2 ijms-23-16074-f002:**
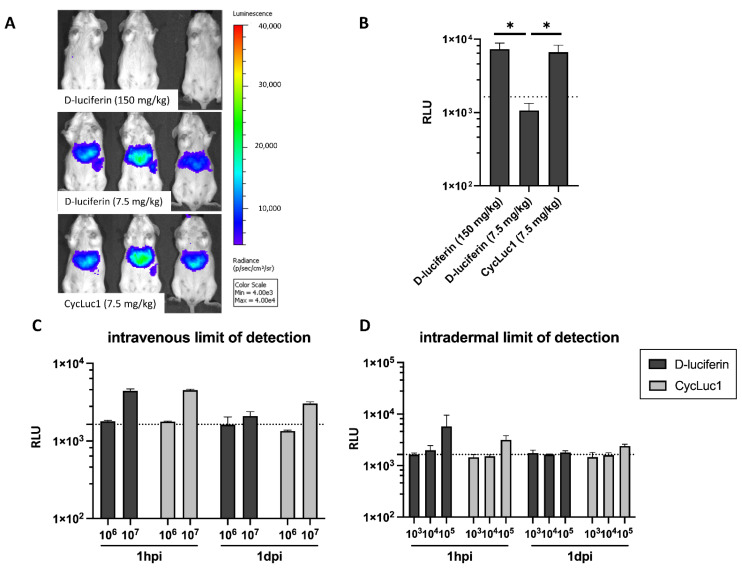
In vivo detection limit of D-luciferin and CycLuc1. (**A**) Bioluminescent pictures of BALB/c mice infected with 10^7^ *L. infantum* LLM-2346*^PpyRE9^* promastigotes after administration of recommended concentrations of D-luciferin (150 mg/kg) and CycLuc1 (7.5 mg/kg) or an equal concentration of D-luciferin (7.5 mg/kg); (**B**) Graphic representation of the bioluminescent output of the mice shown in panel A upon administration of the recommended and equal concentrations of D-luciferin and CycLuc1; In vivo LoD after intravenous (**C**) and intradermal (**D**) infection with *L. infantum* LLM-2346*^PpyRE9^* in BALB/c with D-luciferin (150 mg/kg) or CycLuc1 (7.5 mg/kg) at 1 h post infection (1 hpi) and 1 day post infection (1 dpi). Dotted lines represent the background signal. Results are expressed as mean relative luminescence units (RLU) [p/s/cm*2*/Sr]) ± SD measured at the whole body level (* *p* ≤ 0.05) (*n* = 3).

**Figure 3 ijms-23-16074-f003:**
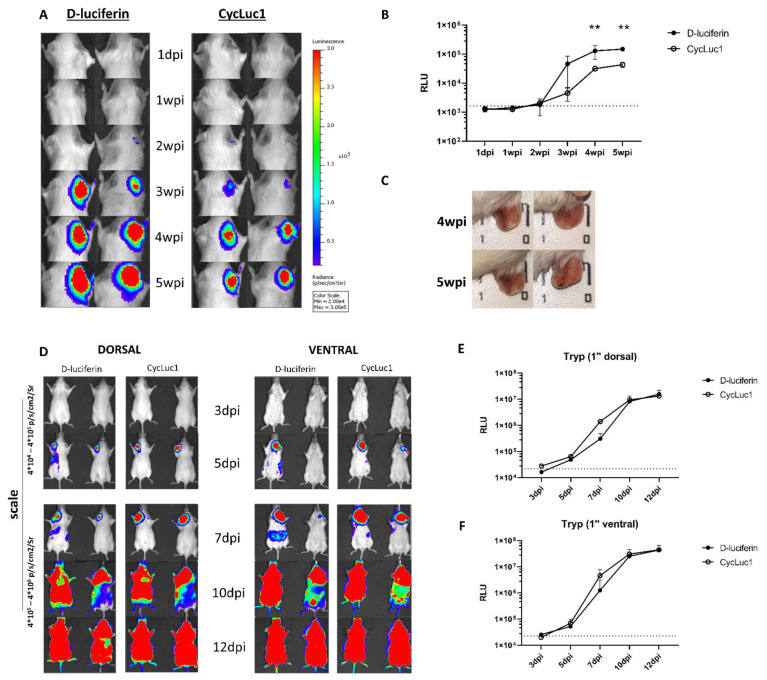
Bioluminescence with D-luciferin and CycLuc1 after sand fly transmission of *L. major* JISH118*^PpyRE9^* and tsetse fly transmission of *T. brucei brucei* AnTAR*^PpyRE9^* in BALB/c or Swiss mice. (**A**) Representative bioluminescent images of BALB/c mice infected with *L. major* JISH118*^PpyRE9^* in the right ear via sand fly transmission after administration of D-luciferin (150 mg/kg) and CycLuc1 (7.5 mg/kg); (**B**) Graphic representation of the bioluminescent signal of panel (**A**). Results are expressed as mean ± SD (*n* = 3); (**C**) Pictures of the ears of BALB/c mice infected with *L. major* JISH118*^PpyRE9^* via sand fly bites showing induced lesions. (**D**) Representative bioluminescent images of Swiss mice infected with *T. brucei brucei* AnTAR*^PpyRE9^* in the left ear via tsetse fly bites after administration of D-luciferin (150 mg/kg) and CycLuc1 (7.5 mg/kg); (**E**,**F**) Graphic representation of the total body bioluminescent output of (**D**). Results are expressed as mean relative luminescence units (RLU) [p/s/cm^2^/Sr]) ± SD (** *p* ≤ 0.01) (*n* = 2). (dpi: days post infection; wpi: weeks post infection).

## Data Availability

The datasets used and/or analyzed during the current study are available from the corresponding author on reasonable request.

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
