# Peer review of "Comparison of Bioluminescent Substrates in Natural Infection Models of Neglected Parasitic Diseases"

_ijms, 2022, doi:10.3390/ijms232416074_

Round 1

Reviewer 1 Report

The present manuscript represents a valuable work to expand not only the bioluminescent reporter palette for in vivo imaging of trypanosomatid infection dynamics but also the comparison of classical vs. novel substrates of the red-shifted luciferase PpyRE9 to i) enhance the light-emitting capability of such enzymes, ii) investigate their differential kinetics/ biodistribution and iii) reduced injection doses.

As an added value, the authors used the always laborious natural transmission routes for both Leishmania spp. and Trypanosoma brucei brucei parasites. However, as for T. brucei the in vitro culture conditions, parasite stages and in vivo experiments lack some information as detailed below.

I recommend the manuscript for publication after minor revision.

- Line 69: AkaLumine-HCl not cited

- Line 96: ethics statement and protocol no. are missing

- Linea 102: the characterization of the transfected parasites is missing. Add it as Supplementary files or quote a reference.In addition, the maintenance of Leishmania spp. virulence/infectivity has not been mentioned.

- Line 108: gene name in italics

- Line 117: hygromycin concentration

- Line 119: include the strain’s pleomorphic capacity

- Line 122: add details on sand fly and tsetse fly maintenance. Better “Confirmation of sand fly and tsetse fly infection for transmission studies”? How many dissected insaects? Infection rate?

- Line 142: T. brucei brucei  “parasites” means, a mixture of both slenders and stumpies? Directly isolated from infected murine blood? Peak of parasitemia? Timing? If not, the medium used to culture T. brucei parasites is missing.

- Line 143: in vitro testing not performed in late stationary-phase promastigotes. It would be helpful since in vivo infections (either experimental or natural transmission experiments) are done with this parasite stage. Also, to compare the differential bioluminescent signal between different parasite stages.

- Line 157: include L. major before the strain’s name.

- Linea 164: (gas) anesthesia is not mentioned. Indicate in the title that only L. infantum was used.

- Linea 171: 15 min of imaging is long enough to damage murine corneas. Please specify whether a protective ointment was used.

- Line 178: “in parallel” means at the same time or separately? Imaging of non-infected mice together with infected animals modifies the real signal background of non-infected controls.

- Line 180: “to evaluate” THE “in vivo application”

- Line 187: define “early” and “late” infection times

- Line 189: again, 10 min of imaging seems quite long for murine corneas. Please indicate animal welfare and/or refinement procedures put in place. 1 second seems rather short.

- Linea 196: include how non-infected controls were used to subtract the signal noise background.

- Line 205: in the light of the in vitro results and the sentence on line 216, I suggest to modify the title of the paragraph as follows: CycLuc1 has a moderately higher potency to detect parasites in vitro

- Lines 209-211:  The description of fig 1 requires refinements. The differences for L. major amastigotes are not statistically significant (Fig. 1D). Instead, differences were seen for L. major promastigotes only at the highest concentration (Fig. 1B). The lowest concentration against L. infantum promastigotes is not statistically significant. Please try to modify accordingly.

- Line 228: I would specify L. infantum in the title of this section.

- Line 232: supplementary figure legends and titles are missing. It is not clear the selected time point for image acquisition in Fig. S1. I would suggest to indicate “1 hpi” or “1 dpi” in the text or on the figure.

- Line 233: is the non-specific hepatic background signal lasting for longer times post infection? Did the authors perform a longer follow-up upon Akalumine-HCl injection?

- Line 240: I would conclude “CycLuc1 can be administered at lower doses in “visceral” leishmaniasis models”

- Line 243: which is the parasite dose injected in A) and B) panels?

- Line 245: whole-body ROI in B), C) and D) panels?

- Figure 3A: please include the bioluminescent signal scale

- Figure 3D: please indicate the scale units

- Figure 3B, E & F: please include a dotted line to indicate the background signal

Reviewer 2 Report

Summary:  Currently, reporter gene technology is a key tool into the transfection models used in the engineering genetic research. In a matter of fact, the brilliant green gen is very common into the molecular expression of recombinant proteins under genetic engineering studies and it  is one of  the greatest contribution to the modern molecular gene research. Here authors display an original research about several bioluminescent reactives in order to use it into in vivo model infection research, and they accomplished it using some neglected parasites of human health importance in thiese kind of infection-model. Is clear that the methodology used by them let to authors to obtain a true knowledge in this interesting area.

Scientist approach concept: The current MS shows an experimental design in order to testing several bioluminescent substrates that is difficult to approach by any researcher do not specialized in this area, obviously like almost all, the current MS has some weakness but this is not impossible to resolve with a detailed reviewing by the authors. The general in vitro and in vivo approach used here to test this three bioluminescent substrates  is enough clear for everyone that want to read this MS, their methodology and analysis of results shows professionalism.

Specific comments:

Title:

Comparison of bioluminescent substrates in natural infection models of neglected parasitic diseases.”

- Usually, in order to get a deep knowledge and interest at first glance an original tittle ought to be extremely communicative, so, a higher communicative title should be written based in material and methods or well in conclusions; both of them are the best manner to write scientist titles in papers. So, the authors ought to choice the appropriate intended words when they are writing a title just based in a comparative description of specific effects between two different bioluminescent substrates. The word: Comparison, on this title is driving to a very general comparative issue and result in a title that is a little poor from original point of view and this does not communicate enough value information about the author´s work. The authors ought to review again carefully their title to be sure that they are describing accurately the scope of research that they have done and they are trying to communicate it to the potential readers. At last, if authors would think in another title, it should be more appropriate in order to communicate in a better way all that they done here, if they cannot it, please kept your title as it was wrote. For example, it could be conclusion-based instead material & methods-based.

Line 6 to 14. Please review it looks a little wrong.

Abstract:

Authors know very well that a good abstract is like a mini-version of the paper, with a clear background, identifying the main scientific question to answer, giving brief results, appropriate interpretation, and last the conclusion, all within a single paragraph. Under this precept I feel that the current MS was wrote accomplishing all goals. Congratulations.

Keywords:

Regularly, in the Keywords section, authors do not should repeat words from your own title Authors made it very well here because almost of them are different to their title, even almost all are narrowly related with the purpose of their research. Only BLI looks like not very common word or term, please review it. For example, you can try with Reporter gene technology or something like that. All remains keywords are OK.

Introduction

This introduction section was written with enough background knowledge, objectivity, arousing great interest and novelty.

Material and Methods

Line 99 to 101, and Line 165, and Line 180.  How the Authors calculate the sample size for the repetition of each experimental treatment used? The authors must to describe how they make this calculation or quoted a suitable reference, because this is key for all subsequent experimental development of their trials.

Line 104 and 105 is dr or Dr

Line 108. Complete physical address for GenScript, or well the authors ought to indicate how another researcher could be able to contact this material, reactive or services supplier. GenScript Biotech (Netherlands) B.V.?

Line 109. Complete physical address for Jena Biosceineces, or well the authors ought to indicate how another researcher could be able to contact this material, reactive or services supplier.

Line 111. Complete physical address for Qiagen, or well the authors ought to indicate how another researcher could be able to contact this material, reactive or services supplier. Since here, please review all remaining paragraphs of your MS in order to accomplish with this issue.

Line 128 to130. This paragraph needs a valid reference.

Line 150. Please indicate what kind of device is the GloMax Explorer, is a spectrophotometer?  Specifically, what is it?

Line 154. If your reference here is as elsewhere, authors need more than just one quote

Line 157. Authors need here a reference for the working dilution used (15:1) or well, authors should explain here how them determined this specific dilution as the better working dilution.

Line 169. Burdens or Burden?

Line 181 to 203. Authors did not verify the normal distribution of residuals and variance homogeneity of the data that they consider for their parametric analysis. Authors did not indicate here clearly what type of variables they analyzed in their 2-way ANOVA analysis, What variable as level and what variable as factor they consider here?. At last, all of these variables are parametric? How they know it? The Mann Whitney test is a comparative test for non-parametric variables but only between an experimental group and their respective unique control group, and do not between three different experimental groups, in this case, Why the authors did not to use a non-parametric test like Kruskal-Wallis?. Please explain why you used here the Mann Whitney test instead the Kruskal-Wallis test. Why authors use at first time in their 2-way ANOVA as pos hoc test the Tukey test and later in the same kind of test they used the Sidak multiple comparison test. What is the main reason to do it?  Why did not use Tukey test for both of them?

Remark: The methodology although clear most of the time, sometimes seems a little imprecise, but with a recurrent review to their MS it could improve significantly. Apparently, the methodology proposed here helped to the authors to pursue and successful conclude their main goal that in this MS is enough clear.

Results

Line 226. This term is not very common in English: in triplo, please verify, you could instead use:  “by thrice times”, at end it is your decision.

Line 229. Please revise syntax for next: in in vivo

Line 240-241. Fig. 2A looks incomplete because authors did not testCycLuc1 at higher dose of 7,5 mg/kg. Please fixed all images in this figures in order to give a better look.

Line 239-240. This is not a result is part of a discussion section, please verify it.

Line 228. Line 251 and 252. These subtitles look as conclusions do not as subtitles. Please verify appropriate syntax for them.

Line 256. Why authors do not check doses higher tha 7.5 mg/kg for CyLuc in this assay, yes seems there a trend but it is lower than the bioluminescence observed with D-luciferin

Line 258. Fig S2 ? Where is it? If this is in the supplementary section, it usually should not quote at the main body of the MS.

Line 258. Fig S3 ? The three images at the supplementary are not very large, so, It could add to the MS instead. It could be more didactic.

Line 279. Only your n=2? I think your statistical degrees of freedom are low n=1.

Discussion

Line 291 to 293. Authors might add something about why they think (with valid references) that all these new red-shifted luciferases and the other modified substrates different to D-luciferin have more capacity to penetrate and spread throughout different cell membranes.

Line 312. is or was? Please review grammar for this discussion section. Usually this section is written in past do not in present time, however, the conclusion usually is written in present time.

Line 318. Please review gramma for this line.

Line 323. Please review gramma for this line.

Line 324. According to your results it is 20-fold lower. 10-fold lower?

Line 330-331. Please review gramma for this line.

Conclusions?

Exactly, there is not here any conclusion, authors ought to formulate a valid conclusion for their study.
